# Fungicidal Activity of Silver and Silica Nanoparticles against *Aspergillus sydowii* Isolated from the Soil in Western Saudi Arabia

**DOI:** 10.3390/microorganisms11010086

**Published:** 2022-12-29

**Authors:** Nuha M. Alhazmi

**Affiliations:** Department of Biology, College of Science, University of Jeddah, Jeddah 21589, Saudi Arabia; nmalhazmi@uj.edu.sa

**Keywords:** silver nanoparticles, silica nanoparticles, *Aspergillus sydowii*, antifugal activity

## Abstract

*Aspergillus sydowii* is a mesophilic soil saprobe that is a food contaminant as well as a human pathogen in immune-compromised patients. The biological fabrication of silica and silver nanoparticles provides advancements over the chemical approach, as it is eco-friendly and cost-effective. In the present study, *Aspergillus sydowii* isolates were collected from the soil fields of six different sites in the western area of Saudi Arabia and then identified using the PCR technique following sequencing analysis by BLAST and phylogenetic analysis. Then, applied silica and silver nanoparticles were synthesized by biological methods, using *Aspergillus niger* as a reducer. Silver and silica nanoparticles were characterized by X-ray diffraction (XRD), scanning electron microscopy (SEM), and transmission electron microscopy (TEM). The antifungal activity of silver and silica nanoparticles against *Aspergillus sydowii* isolates was evaluated using the disc diffusion method and the minimum inhibitory concentration (MIC). The physiochemical results emphasized the fabrication of silver and silica nanoparticles in spherical shapes with a diameter in the range of 15 and 40 nm, respectively, without any aggregation. MIC of Ag-NPs and Si-NPs against *Aspergillus sydowii* isolates were 31.25 and 62.5 µg/mL, respectively. Finally, the aim of the study is the use of silver as well as silica nanoparticles as antifungal agents against *Aspergillus sydowii*.

## 1. Introduction 

The soil disease aspergillosis causes significant deterioration and partial massive mortalities in the soil ecosystem [1]. The first study reported that the ascomycetous fungus *Aspergillus sydowii* caused disease in gorgonians in 1995 [2]. Also, *Aspergillus sydowii occurs* in diverse soil and marine ecosystems [3]. Moreover, *Aspergillus sydowii* is a mesophilic soil saprobe that is a food contaminant as well as a human pathogen in immune-compromised patients [4]. Furthermore, the pathogenicity of *Aspergillus sydowii* depends on the genetic composition and immune status of the host [5]. Also, *Aspergillus sydowii* causes devastating aspergillosis in coral reefs but is a very rare cause of human disease [6]. Nanoscience is a multidisciplinary science interested in the unique properties of nanoparticles. It is defined as a transformation of bulk material into nanosize (1–100 nm). Moreover, nanoscience has various applications in different fields like industry and agriculture [7]. Currently, scientists focus on the benefits of nanomaterial properties such as large surface area and optical and chemical properties such as noble metals and metal oxides [8]. Silver nanoparticles are promising materials in the fields of antibacterial, antitumor, and antiviral therapy [9,10]. Interestingly, Ag-NPs have a broad spectrum of antibacterial activity [10]. The activity of silver nanoparticles as antibacterial agents depends on their ability to attack microorganisms in multiple structures at a time and kill many types of bacteria [11]. Silica nanoparticles have many applications in medical science, such as drug delivery, gene therapy, diagnosis, and imaging. Due to their unique properties, the Nanotechnology Consumer Products Inventory (NCPI) in 2015 classified SiNPs as among the top five nanomaterials of consumer products. Interestingly, soluble silicon can work as a modulator of host resistance to pathogens. The central role of silica nanoparticles may be feasible to enhance fungal resistance in maize which is one of the major food crops in the world. In this study, *Aspergillus sydowii* fungus was isolated from 100 samples of soil collected from different areas in the western area of Saudi Arabia. Silver nanoparticles and silica nanoparticles, which were synthesized using *Aspergillus niger*, were used as reducers, and we investigated their fungicide activity against *Aspergillus sydowii*.

## 2. Material and Methods 

### 2.1. Sampling 

The soil samples (100 samples) were collected from the soil of 6 different sites in the western area of Saudi Arabia and then preserved in sterile zipper plastic bags with collection place and sample number labeling, as shown in Table 1. Then transfer the samples to the laboratory until they are needed.

### 2.2. Fungal Isolation

The fungi were isolated from the subsurface layer of soil (ca. 15–30 cm) by using the soil plate method, according to Warcup, 1950 [11]. In brief, 0.005 g of soil was spread in the bottom of a sterile Petri dish and then cooled (40–45 °C). Potato Dextrose Agar (PDA) mediums (Pronadisa, Madrid, Spain) were added. The plates were incubated at 28 ± 2 °C for 4–5 days. Three plates were prepared from each soil sample, and each morphologically unique fungal colony was subcultured and purified. The purified fungal isolates were stored at 4 °C on a PDA slant. Then, the stock cultures were maintained in the refrigerator, and subcultures were carried out at monthly intervals. Fresh cultures were prepared at four-month intervals. by streaking onto PDA plates from stock slants to check for purity and then sub-culturing on fresh PDA slants. Plates and slants were both incubated at 28.2 °C. 

### 2.3. DNA Extraction, PCR Amplification, Sequencing, and Species Identification

DNA was extracted from 20 mg of the fungal isolate tissues using the DNeasy extraction kit (Qiagen, Inc., Venlo, The Netherlands) according to the manufacturer’s protocol. For DNA fragments containing internal transcribed spacers, the fragments fungi primer is (Forward (ITS1): TCCGTAGGTGAACCTGC, Reverse (ITS4): TCCTCCGCTTATTGATATGC). Including 5.8S, were amplified and sequenced with primer pair ITS5/ITS4. The PCR profile was: 2.5 μL 10× buffer, 1.4 μL 50 mM MgCl_2_, 1.6 μL 25 mM dNTPs, 0.5 μL of each 10 mM primer (forward and reverse), 1 μL 1 mg/mL BSA, 1 μL DNA, 0.3 μL 5 U/μL Taq polymerase, and 16.2 μL ddH_2_O. The PCR conditions were 1 min at 95 °C, 35 cycles of 1 min at 95 °C, 45 s at 58 °C and 1 min at 72 °C, and finally, 10 min at 72 °C. The amplicons were sequenced for both strands using Big Dye Terminator in an ABI 3730 genetic analyzer (Applied Biosystems, Waltham, MA, USA). The sequences were edited, and primers were trimmed using Seq Studio (Life Technologies, Carlsbad, CA, USA) and run following a medium module. BLAST was used to compare the sequences against those existing in the National Center of Biotechnology Information (NCBI) nucleotide databases. Sequencing was performed using Seq Studio (Life Technologies, Carlsbad, CA, USA) and run following a medium module [12,13].

### 2.4. Synthesis of Silver Nanoparticles Using Aspergillus niger

Silver nanoparticles were synthesized using the biological method due to the toxicity of the chemical reducers, such as sodium borohydride. Interestingly, silver ions were reduced on the surface of fungal microbes. In brief, silver nanoparticles were synthesized by using *Aspergillus niger* RCMB 002F008. Five hundred ppm of AgNO_3_ dissolved in the fungal culture (MGYP broth medium), including malt (3 g), glucose (10 g), yeast extract (3 g), and peptone (5 g). The pH of the media was adjusted at 6.2 ± 0.2. The fungal cultures were grown while aerobically agitated at 28 °C and the incubator was shaken at 150 rpm for 7 days. Then, the sterile filtration and re-suspension in sterile deionized distilled water were achieved. The supernatant was kept at ±4.0 °C in a refrigerator.

### 2.5. Synthesis of Silica Nanoparticles Using Aspergillus niger

Silica nanoparticles were synthesized by using *Aspergillus niger* (RCMB 002F008). 500 ppm of Na_2_ SiO_3_ dissolved in the fungal culture (MGYP broth media). The pH of the media was adjusted at 6.2 ± 0.2. The fungal cultures were grown aerobically agitated at 28 °C and the incubator was shaken at 200 rpm for 5 days. Then, the sterile filtration and re-suspension in sterile deionized distilled water were achieved. The supernatant was kept at ±4.0 °C.

### 2.6. Characterization of Silver and Silica Nanoparticle

The characterization of the silver nanoparticles by the following physiochemical techniques by the Fourier transformed infrared (FT-IR) spectrum via the Nicolet 6700 apparatus (Thermo Scientific Inc., Waltham, MA, USA), The crystalline nature and grain size were analyzed b XRD (D8 Advance X-ray diffractometer, Bruker, Germany), the morphology and visualization of zinc oxide nanoparticles were evaluated using transmission electron microscopy (TEM; JSM-2100F, JEOL Inc., Tokyo, Japan) and scanning electron microscopy (SEM; JSM-690, JEOL Inc., Tokyo, Japan).

### 2.7. Antifungal Disk Diffusion Method

The *Aspergillus sydowii* strains were cultured on Sabouraud dextrose agar and then incubated at 35 °C for 24 h. Also, for the mold fungi, the fungal strains were cultured on a potato dextrose agar slant and incubated at 35 °C for 5 days. Using a sterile loop, pure colonies of the *Aspergillus sydowii* species were transferred into a tube containing sterile normal saline. For the mold, 1 mL of sterile distilled water supplemented with 0.1% Tween 20 was used to cover and resuspend the colonies. Using a hemocytometer, the suspension was adjusted to 2–5 × 10^6^ conidia/mL. The suspension was further diluted at 1:10 to obtain the final working inoculums of 2–5 × 10^5^ conidia/mL. The inoculums were poured over MHA supplemented with 2% of glucose. The sterile 6 mm disks that were impregnated with 20 μL test compound (with a concentration of 10 µg/mL) were placed over the plate. The standard antifungal drug Nystatin was used as a positive control, with sterile distilled water as a negative control, and incubated at 35 °C for 48 h. The zone of inhibition was measured in millimeters [14].

### 2.8. The Susceptibility of Aspergillus sydowii Isolates

The MICs for *Aspergillus sydowii* were determined using the reference procedure of the Antifungal Susceptibility Testing of CLSI M27-A3 and EUCAST for the testing of fermentative yeasts. MICs for *Aspergillus sydowii* were determined in accordance with EUCAST and CLSI M38-A. Briefly, testing was performed in sterile 96-well microtiter plates with Roswell Park Memorial Institute (RPMI) 1640 medium with l-glutamine, without sodium bicarbonate (NaHCO_3_, RPMI 1640; Gibco, Carlsbad, CA, USA) supplemented with 2% glucose, buffered to pH 7.0 with 4-(2-hydroxyethyl)-1-piperazineethanesulfonic acid (HEPES) medium [15,16,17].

### 2.9. Minimum Inhibition Concentration (MIC) Assay 

Regarding the preparation of yeast inoculums, the fungal strains were subcultured on a Sabouraud’s dextrose agar slant and incubated for 24–48 h at 35 °C to obtain a freshly grown pure culture. The homogenous suspension was adjusted to 0.5 McFarland standards. Then, the inoculum size was further adjusted to 0.5 × 10^5^ or 2.5 × 10^5^. In addition, the mold suspension of conidia was obtained from 5 days of culture on a Sabor and dextrose agar slant incubated at 35 °C. Colonies were covered with 5 mL of sterile distilled water supplemented with Tween 20. The conidia were collected with a sterile cotton swab, transferred to a sterile tube, and vortexed to homogenize the suspension. The suspension was standardized by counting the conidia in a hemocytometer to 2–5 × 10^6^ conidia/mL. The suspension was diluted 1:10 with RPMI to obtain final inoculums of 2–5 × 10^5^ conidia/mL. A total of 50 μL of each compound with a concentration of 1 mg/mL concentration and 50 μL of fungal suspension were added to each well for the negative control lane, but 100 μL of broth was added to the positive control lane (each well reached the final desired concentration of 2–5 ×10^5^ CFU/mL). The plate was sealed with aluminum foil and incubated at 35 °C for 24–48 h in a humid atmosphere. The MIC was determined using an ELISA reader at 530 nm for the yeast species and visually for mold species after 48 h of incubation as the lowest concentration of drug that resulted in 50% inhibition of growth compared to the drug-free growth control [18]. 

## 3. Results 

### 3.1. Fungal Isolation and Species Identification

The phylogenetic analyses resulted in all the isolated phylogenetically supported clusters (Figure 1). The majority of the isolates could be assigned to the species *Aspergillus sydowii*, which was submitted to GenBank and had an accession number (GenBank: ON887208.1).

### 3.2. Characterization of Silver Nanoparticle 

This study’s goal was to investigate the antibacterial effect of synthesized silver nanoparticles. The synthesized silver nanoparticles used in this study were characterized by X-ray diffraction (XRD) and TEM. The XRD patterns for synthesized silver nanoparticles demonstrated showed that five main characteristic diffraction peaks for Ag were observed at 2θ = 35.634, 43.415, and 65.264°, which are assigned to 111, 220, and 400, respectively. It was matched with the standard silver diffraction pattern according to the Joint Committee on Powder Diffraction standards (JCPDS-4-0783 Diff. card), as Figure 2 shows. As shown in Figure 3, the SEM images depicted the spherical shapes of silver nanoparticles with smooth surfaces. The TEM image displayed the silver nanoparticle with particle sizes of 14–16 nm, as shown in Figure 4, emphasizing the fabrication of silver nanoparticles of regular shapes.

### 3.3. Characterization of Silica Nanoparticle 

This study’s goal was to investigate the antibacterial effect of synthesized silica nanoparticles. The synthesized silver nanoparticles used in this study were characterized by X-ray diffraction (XRD) and TEM. The XRD patterns for synthesized silica nanoparticles demonstrated showed that the main characteristic diffraction peaks for Si were observed at 2θ = 35.634°, which were assigned to 111. It was matched with the standard silver diffraction pattern according to the Joint Committee on Powder Diffraction standards (JCPDS-16-1157 Diff. card), as Figure 5 shows. As shown in Figure 6, the SEM images depicted the spherical shapes of silica nanoparticles with smooth surfaces. The TEM image displayed the silica nanoparticles with particle sizes of 40 nm, as shown in Figure 7, emphasizing the fabrication of silica nanoparticles of regular shapes.

### 3.4. The Susceptibility of A. sydowii Isolates against Silver and Silica Nanoparticles

Silver and silica nanoparticles have good anti-fungal activity against many fungi, such as *Candida albicans*, *Aspergillus niger*, and *Pestalotiopsis maculans* [10]. As Table 2 and Figure 8 show, silver nanoparticles synthesized by the reduction of silver ions by *Aspergillus niger* has a strong effect against *Aspergillus Sydowii* isolates. The inhibition zone of *Aspergillus sydowii* isolated from the soil was 35 nm. Also, silica nanoparticles have less effect on *Aspergillus sydowii* isolated from the soil, and the inhibition zone was 27 nm. Similarly, the minimal inhibition concentration of silver nanoparticles against *Aspergillus sydowii* isolated from the soil is better than silica nanoparticles against *Aspergillus sydowii* (28.3 and 62.5 µg, respectively).

## 4. Discussion

*Aspergillus sydowii* is a mesophilic soil saprobe. It is a food contaminant as well as a human pathogen. The pathogenicity of *Aspergillus sydowii* depends on the genetic composition and immune status of the host, as well as the time of exposure [19]. For example, *Aspergillus fumigatus* may cause aspergillosis, which is fatal to immune-compromised humans. However, it is less common in healthy people [20]. The *A. sydowii* growth rates depend on the temperature, which promotes the emergence and pathogenicity of aspergillosis. The reduction from the higher temperature may be due to the inactivation of antifungal compounds and increased fungal resistance. The *A. sydowii* growth rates depend on the temperature, which promotes the emergence and pathogenicity of *aspergillosis*. The reduction from the higher temperature may be due to the inactivation of antifungal compounds; increased fungal resistance. The antifungal compounds are not inactivated at higher temperatures. Thus, the effect of high temperatures appears to be to promote the growth of the fungus, allowing it to overcome the host’s defenses [21]. Nanoscience is interesting in the fabrication and reshaping of the material and in transforming the bulk material into the nanoscale [22,23]. Nanoscience contributes to the development of different materials to enhance plant growth. Silica (SiO_2_) is an essential element for monocot plants, and it is known to confer biotic and abiotic stress tolerance [24,25]. Interestingly, silver and silica nanoparticles have a broad spectrum of antibacterial activity [10]. In this study, we collected soil samples from various areas in Jeddah, Saudi Arabia’s western region, then isolated *Aspergillus sydowii* from the soil using the Warcup method and identified the fungal species using the PCR technique after sequencing analysis using BLAST and phylogenetic analysis. Then, we synthesized silver and silica nanoparticles using a biological approach by applying *Aspergillus niger, which* worked as a reducing agent and characterized the silica and silver nanoparticles using XRD, SEM, and TEM techniques. We examine the antifungal activity of both silver and silica nanoparticles against *Aspergillus sydowii* using the antifungal disk diffusion method and MIC assay. The results displayed that the phylogenetic analyses of the majority of the isolates could be assigned to the species *Aspergillus sydowii*. The physiochemical characterization of the silver and silica nanoparticles had particle sizes of 14 and 40 nm, respectively. The inhibition zone of *Aspergillus sydowii* was 26 nm and 21 nm against silver and silica nanoparticles, respectively. The minimum inhibitory concentration (MIC) of *Aspergillus sydowii* was 31.25 µg and 62.5 µg for silver and silica, respectively. The antimicrobial activity and surface-to-volume ratio increased as the size of the nanoparticle decreased [26,27]. Noble metals like silver and silica have high antibacterial activity while being non-toxic to animal cells [28]. Silica nanoparticle accumulation enhances leaf erectness, which leads to improved defense against fungal pathogens. Hence, nanosilica acts as an effective physical barrier against mycelial invasion. Silica nanoparticles can impact enzyme activities such as PPOs, peroxidases, and PALs, which increases the resistivity of maize roots [29]. Furthermore, silica nanoparticles induce the expression of enzymes as well as fungal infections [29]. The antimicrobial mechanism of the silver nanoparticle is the release of silver ions throughout the accumulation of extracellular Ag-NPs, which activate the penetration of Ag^+^ inside the cell [10]. Also, the attachment of Ag-NPs to DNA and thiol groups of proteins with rectification with phosphorus- or sulfur-containing compounds inside DNA may lead to damage in the yeasts by inhibiting DNA replication and protein inactivation [30]. Finally, silver and silica nanoparticles are recommended as antifungal agents against *Aspergillus sydowii*.

## 5. Conclusions

*Aspergillus sydowii* is a mesophilic soil saprobe. It is a food contaminant as well as a human pathogen. The pathogenicity of *Aspergillus sydowii* depends on the genetic composition and immune status of the host, as well as the time of exposure. Nanoscience is a multidisciplinary science with interest in the unique properties of nanoparticles. In the current study, *Aspergillus sydowii* isolates were collected from soil fields in Saudi Arabia’s western region and identified using the PCR technique after sequencing analysis using BLAST and phylogenetic analysis. Then the applied silica and silver nanoparticles were synthesized by biological methods, using *Aspergillus niger* as a reducer. Silver and silica nanoparticles were characterized by X-ray diffraction (XRD), scanning electron microscopy (SEM), and transmission electron microscopy (TEM). The antifungal activity of silver and silica nanoparticles against *Aspergillus sydowii* isolates was evaluated using the disc diffusion method and minimum inhibitory concentration (MIC). The physiochemical results emphasized the fabrication of silver and silica nanoparticles in spherical shapes with a diameter in the range of 15 and 40 nm, respectively, without any aggregation. The MIC of Ag-NPs and Si-NPs against *Aspergillus sydowii* isolates were 31.25 and 62.5 µg/mL, respectively. Finally, silver and silica nanoparticles are recommended as antifungal agents against *Aspergillus sydowii*.

## Figures and Tables

**Figure 1 microorganisms-11-00086-f001:**
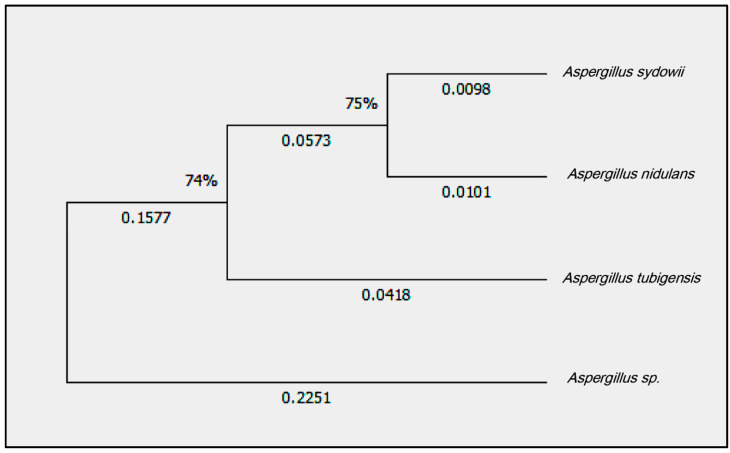
Phylogenetic tree of the fungal isolate *Aspergillus sydowii* and some related fungal isolates from the database (*Aspergillus nidulans*), *Aspergillus tubigensis*, and *Aspergillus* sp.

**Figure 2 microorganisms-11-00086-f002:**
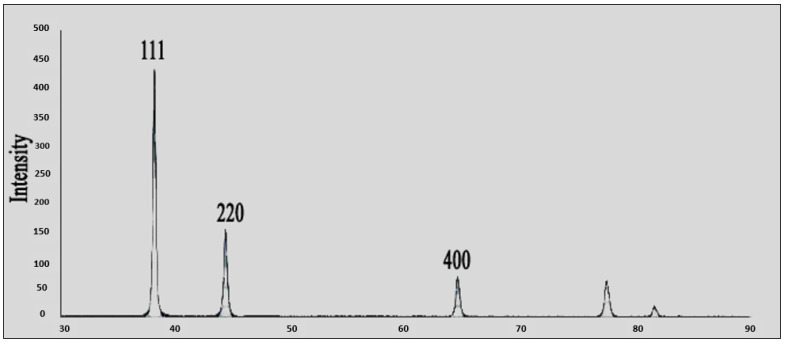
XRD of silver nanoparticles.

**Figure 3 microorganisms-11-00086-f003:**
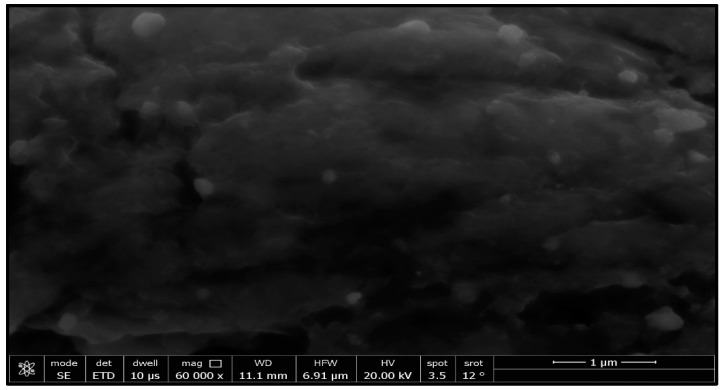
SEM images of silver nanoparticles.

**Figure 4 microorganisms-11-00086-f004:**
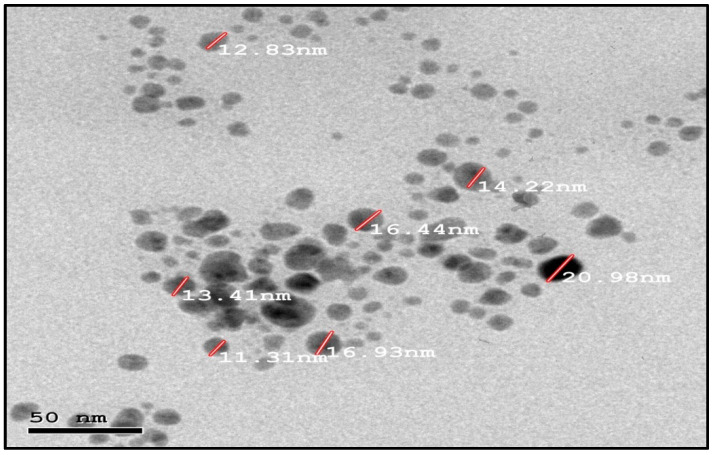
TEM images of silver nanoparticles.

**Figure 5 microorganisms-11-00086-f005:**
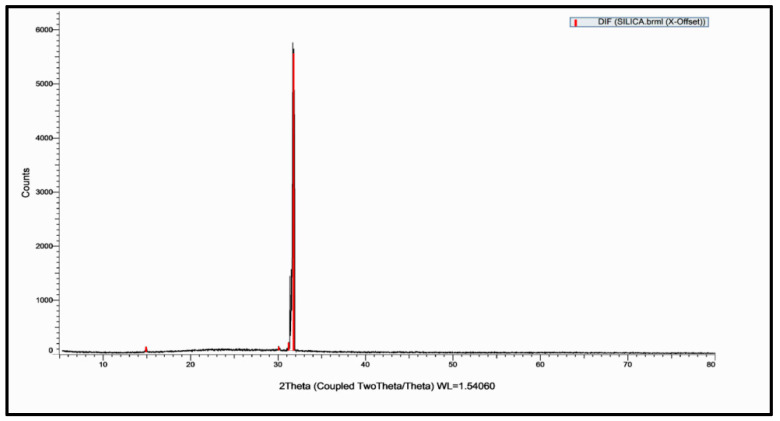
XRD of silica nanoparticles.

**Figure 6 microorganisms-11-00086-f006:**
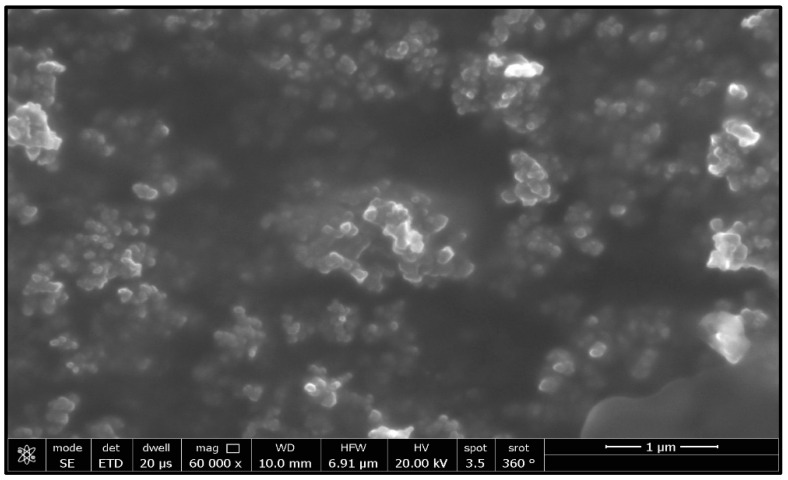
SEM images of silica nanoparticles.

**Figure 7 microorganisms-11-00086-f007:**
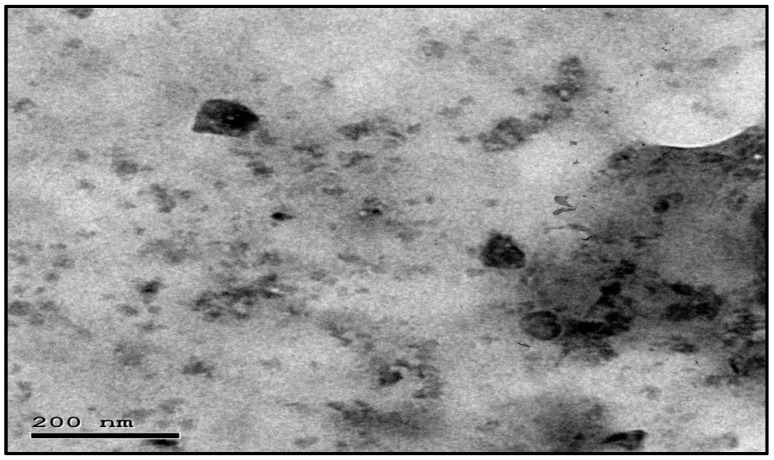
TEM images of silica nanoparticles.

**Figure 8 microorganisms-11-00086-f008:**
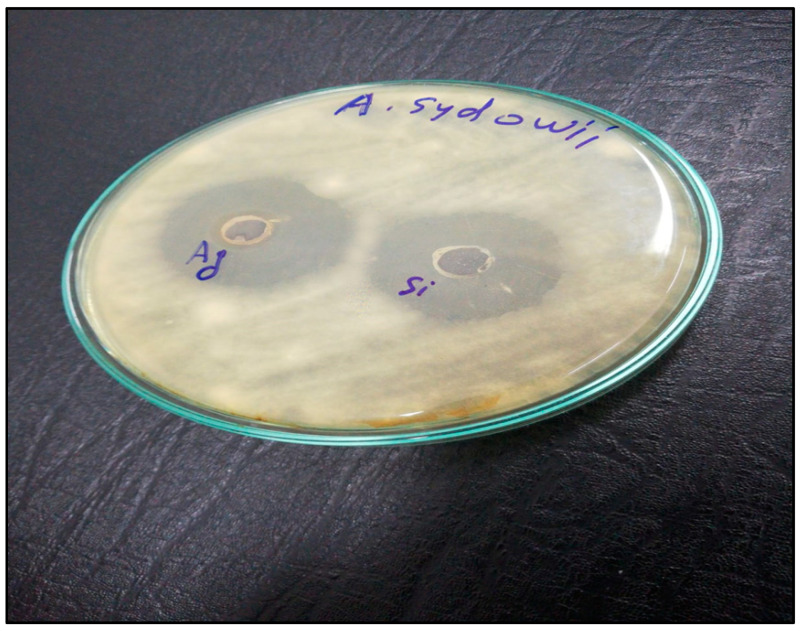
The antifungal activity of silver and silica nanoparticles against *Aspergillus sydowii* isolated from the soil.

**Table 1 microorganisms-11-00086-t001:** Soil samples collected from different places in Jeddah.

Soil Sample Number	Location	The Location Coordinates with GPS
1	Jeddah	21°37′36.3″ N 39°17′56.2″ E
2	Bahrah	21°24′22.3″ N 39°28′40.6″ E
3	Al-Jumum	21°37′08.2″ N 39°57′26.5″ E
4	Taif 1	21°03′46.0″ N 40°26′02.3″ E
5	Taif 2	21°13′31.8″ N 40°25′27.2″ E
6	Mesaan Al Atta	20°42′55.6″ N 40°52′51.2″ E

**Table 2 microorganisms-11-00086-t002:** The antifungal activity of silver and silica nanoparticles against *Aspergillus sydowii* isolated from the soil.

Tested Material	Inhibition Zone (nm) of *A. Sydowii*	MIC (µg) of *A. Sydowii*
Silver nanoparticles	30 nm	31.25 ± 2.1
Silica nanoparticles	27 nm	62.5 ± 1.2

## Data Availability

The original contributions presented in the study are included in the article; further inquiries can be directed to the corresponding author.

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
