# Peer review of "Fungicidal Activity of Silver and Silica Nanoparticles against Aspergillus sydowii Isolated from the Soil in Western Saudi Arabia"

_microorganisms, 2022, doi:10.3390/microorganisms11010086_

Round 1
Reviewer 1 Report
This manuscript deals with the results of "Fungicidal Activity of Silver and Silica Nanoparticles against Aspergillus sydowii Isolated from the Soil in Western Saudi Arabia". The results showed in this manuscript are interesting in academic, although this study might be more deeply to show some mechanisms of Silver and Silica Nanoparticles against Aspergillus sydowii. And there are some writing mistakes inside. However, it is basically qualified for publication in your Journal after author s revise seriously.
There are some suggestions for revision:
Line 11, 260, "the Pcr" should be "the PCR".
Line 25-26, 27, "Aspergillus sydowii" should be in Italic, "Aspergillus sydowii".
Line 58, (40-45 oC) should be written as (40-45 oC).
Line 89, "the surface of fungal of microbial surface." should be changed as "the
surface of fungal microbe."
Line 89, "the media" should be as "the medium".
In text, "µl" was used in some where,but "µL" used. One of which should be choiced from both to be used, that is onle one may be used.
Line 2, 152, 155, 192, 196, 197, 198, 199, 200, 205, 233, 235, 237, 253, 271, and in Table 2"Aspergillus Sydowii" should be as "Aspergillus sydowii".
Line 156,." Aspergillus sp" should be as Aspergillus sp."
Line 194, " Candida Albicans" should be as " Candida albicans".
Line 217 and other places in text, "aspergillosis" should be as "aspergillosis".
Line 230, "which" , Line 234, "the"should be not in Italic.
Line 259, "Aspergillus sydowii" should be in Italic.
The listing styles of References are not uniformed, in some of which the first letter are capital,but in some nof which are not, suggesting they are uniformed as the rules of Journal.

Author Response
POINTWISE RESPONSE TO REVIEWER’s COMMENTS ON Mircoorganisms–MDPI
Dear Mircoorganisms –MDPI Editor, Dear Reviewers,
Thank you for the Reviewer comments for Mircoorganisms –MDPI; they are very helpful to improve our manuscript. Please find attached the revised article entitled “Fungicidal Activity of Silver and Silica Nanoparticles against Aspergillus sydowii Isolated from the Soil in western Saudi Arabia”, for consideration for publication in Mircoorganisms- MDPI.
Each Reviewer comment is indicated below, together with the responses (clarifications/ revisions) i have included in the revision.
It is hoped that this revision satisfies the queries raised by respected Reviewers and manuscript will be considered for publication in Mircoorganisms - MDPI.
With kind regards
Dr. Nuha M. Alhazmi
Corresponding author
Reviewer -1
Line 11, 260, "the Pcr" should be "the PCR".
AUTHORS: Thank you very much for your valuable suggestion. Thanks for Reviewer’s comments; i fixed all the errors.
Line 25-26, 27, "Aspergillus sydowii" should be in Italic, "Aspergillus sydowii".
AUTHORS: Thank you very much for your valuable suggestion. Thanks for Reviewer’s comments; i fixed all the errors.
Line 58, (40-45 oC) should be written as (40-45 oC).
AUTHORS: Thank you very much for your valuable suggestion. Thanks for Reviewer’s comments; i fixed all the errors.
Line 89, "the surface of fungal of microbial surface." should be changed as "the
surface of fungal microbe."
AUTHORS: Thank you very much for your valuable suggestion. Thanks for Reviewer’s comments; i fixed all the errors.
Line 89, "the media" should be as "the medium".
AUTHORS: Thank you very much for your valuable suggestion. Thanks for Reviewer’s comments; i fixed all the errors.
In text, "µl" was used in some where,but "µL" used. One of which should be choiced from both to be used, that is onle one may be used.
AUTHORS: Thank you very much for your valuable suggestion. Thanks for Reviewer’s comments; i fixed all the errors.
Line 2, 152, 155, 192, 196, 197, 198, 199, 200, 205, 233, 235, 237, 253, 271, and in Table 2"Aspergillus Sydowii" should be as "Aspergillus sydowii".
AUTHORS: Thank you very much for your valuable suggestion. Thanks for Reviewer’s comments; i fixed all the errors.
Line 156,." Aspergillus sp" should be as Aspergillus sp."
AUTHORS: Thank you very much for your valuable suggestion. Thanks for Reviewer’s comments; i fixed all the errors.
Line 194, " Candida Albicans" should be as " Candida albicans".
AUTHORS: Thank you very much for your valuable suggestion. Thanks for Reviewer’s comments; i fixed all the errors.
Line 217 and other places in text, "aspergillosis" should be as "aspergillosis".
Line 230, "which" , Line 234, "the"should be not in Italic.
AUTHORS: Thank you very much for your valuable suggestion. Thanks for Reviewer’s comments; i fixed all the errors.
Line 259, "Aspergillus sydowii" should be in Italic.
AUTHORS: Thank you very much for your valuable suggestion. Thanks for Reviewer’s comments; i fixed all the errors.
The listing styles of References are not uniformed, in some of which the first letter are capital,but in some nof which are not, suggesting they are uniformed as the rules of Journal.
AUTHORS: Thank you very much for your valuable suggestion. Thanks for Reviewer’s comments; i modified all the references
Reviewer 2 Report
1. The introduction section has less information about the study; add more information with relevance to this study including recent references.
2. The scientific name of the fungus is written wrongly throughout the manuscript, check and corrects it (First letter should be capitalized in generic name and species name should start with small letter, both generic and species name should be italic, For eg. Aspergillus sydowii).
3. In 2.4 of materials and methods, Line 88 please include the units of weight in case Malt 3…, Glucose 10..Yeast extract 3.., peptone 5..., (If you do measurement of these chemicals).
4. In line 92 and 98, please indicate where you kept the supernatant at 4oC.
5. In line 110, the sentence is not clearly explained, rewrite it.
6. Line 165 As figures 3, when you start with remaining sentence, use small letter first because you already started the sentence with capital letter (check and correct it in the manuscript throughout).
7. Please add the title of Table 2, which you want to represent in the table.
8. In Line 45, Fungus not funguses (Plural- Fungi).
9. In Line 134, correct the 105 as 105.
10. In Line 175, correct the nanoparticle as nanoparticles.
11. In Line 178, use full stop (.) after sentence ending.
12. In Line 179, correct the si as Si.
13. In Line 205, correct antifungals as antifungal.
14. In Line 252 and 253, the text format is not uniform as per journal format.
Author Response
POINTWISE RESPONSE TO REVIEWERS’S COMMENTS ON Mircoorganisms–MDPI
Dear Mircoorganisms –MDPI Editor, Dear Reviewers,
Thank you for the reviewer comments for Mircoorganisms –MDPI; they are very helpful to improve our manuscript. Please find attached the revised article entitled “Fungicidal Activity of Silver and Silica Nanoparticles against Aspergillus sydowii Isolated from the Soil in Western Saudi Arabia”, for consideration for publication in Mircoorganisms- MDPI.
Each reviewer comment is indicated below, together with the responses (clarifications/ revisions) we have included in the revision.
It is hoped that this revision satisfies the queries raised by respected reviewers and manuscript will be considered for publication in Mircoorganisms - MDPI.
With kind regards
Dr. Nuha M. Alhazmi
Corresponding author
Reviewer-2
The introduction section has less information about the study; add more information with relevance to this study including recent references
AUTHORS: Thank you very much for your valuable suggestion. Thanks for reviewer’s comments; I modified the introduction and add recent references. Also, I mentioned more details about the study in the end of the introduction section
The soil disease aspergillosis has a significant deterioration and partial massive mortalities of the soil ecosystem [1]. The first study reported that the ascomycetous fungus Aspergillus sydowii caused disease in gorgonians since 1995[2]. Also, Aspergillus sydowii is ocuured in different diversity such as soil and marine ecosystems [3]. Moreover, Aspergillus sydowii is a mesophilic soil saprobe caused a food contaminant as well as human pathogen in immune-compromised patients [4]. Furthermore, the pathogenicity of Aspergillus sydowii depends on the genetic composition and immune status of the host [5]. Also, Aspergillus sydowii causes devastating aspergillosis in coral Reefs but very rare cause of human disease [6].Nanoscience is a multidisciplinary science interest in the unique properties of nanoparticles. It’s defined as transform of the bulk material into nanosize (1-100 nm). Moreover, nanoscience has variety applications in different field like industry and agriculture [7]. Currently, scientists focus on the benefits of nanomaterial properties such as large surface are, optical and chemical properties such as noble metal and metal oxide metal [8].Silver nanoparticles are promising materials in the fields of antibacterial, antitumor, and antiviral therapy [9-10]. Interestingly, Ag-NPs have a broad spectrum of antibacterial activity [10]. The activity of silver nanoparticles as antibacterial agents depends on ability to attack microorganisms in multiple structures at a time and kill many types of bacteria [11].Silica nanoparticles have many applications in medical science such as drug delivery, gene therapy and diagnosis and imaging. Due to their unique properties, the Nanotechnology Consumer Products Inventory (NCPI) in 2015 classifies SiNPs as the top five nanomaterials of consumer products. Interestingly, the soluble silicon can work as a modulator of host resistance to pathogens. The central role of silica nanoparticles may be feasible to enhance fungal resistance in maize which is one of the major food crops in the world. In this study, Aspergillus sydowii fungus were isolated from 100 samples from soil collected from different area in western area in Saudi Arabia and, utilized silver nanoparticles and silica nanoparticle which synthesized using Aspergillus niger as reducers and investigated their fungicide activity against Aspergillus sydowii.
The scientific name of the fungus is written wrongly throughout the manuscript, check and corrects it (First letter should be capitalized in generic name and species name should start with small letter, both generic and species name should be italic, For eg. Aspergillus sydowii).
AUTHORS: Thank you very much for your valuable suggestion. Thanks for reviewer’s comments; we fixed all the errors.
- In 2.4 of materials and methods, Line 88 please include the units of weight in case Malt 3…, Glucose 10..Yeast extract 3.., peptone 5..., (If you do measurement of these chemicals).
AUTHORS: Thank you very much for your valuable suggestion. Thanks for reviewer’s comments; the unit is gram (g).
“(MGYP broth medium) including Malt 3g, Glucose 10g, yeast extract 3g and peptone 5g”.
In line 92 and 98, please indicate where you kept the supernatant at 4oC.
AUTHORS: Thank you very much for your valuable suggestion. Thanks for reviewer’s comments; in in refrigerator.
In line 110, the sentence is not clearly explained, rewrite it.
AUTHORS: Thank you very much for your valuable suggestion. Thanks for reviewer’s comments; I rewrite it “The Aspergillus sydowii strains were cultured on Sabouraud dextrose agar, and then it’s incubated at 35°C for 24 hours. Also, for the mold fungi, the fungal strains were cultured on were cultured on potato dextrose agar slant and incubated at 35°C for 5 days”.
Please add the title of Table 2, which you want to represent in the table.
AUTHORS: Thank you very much for your valuable suggestion. Thanks for reviewer’s comments; the title of the table 2 is “Table 2 : The antifungal activity of silver and silica nanoparticles against Aspergillus sydowii isolated from the soil”.
Line 165 As figures 3, when you start with remaining sentence, use small letter first because you already started the sentence with capital letter (check and correct it in the manuscript throughout).
AUTHORS: Thank you very much for your valuable suggestion. Thanks for reviewer’s comments; i fixed all the errors.
In Line 45, Fungus not funguses (Plural- Fungi).
AUTHORS: Thank you very much for your valuable suggestion. Thanks for reviewer’s comments; i fixed all the errors.
- In Line 134, correct the 105 as 105.
AUTHORS: Thank you very much for your valuable suggestion. Thanks for reviewer’s comments; i fixed all the errors.
- In Line 175, correct the nanoparticle as nanoparticles.
AUTHORS: Thank you very much for your valuable suggestion. Thanks for reviewer’s comments; i fixed all the errors.
- In Line 178, use full stop (.) after sentence ending.
AUTHORS: Thank you very much for your valuable suggestion. Thanks for reviewer’s comments; i fixed all the errors.
- In Line 179, correct the si as Si.
AUTHORS: Thank you very much for your valuable suggestion. Thanks for reviewer’s comments; i fixed all the errors.
- In Line 205, correct antifungals as antifungal.
AUTHORS: Thank you very much for your valuable suggestion. Thanks for reviewer’s comments; i fixed all the errors.
In Line 252 and 253, the text format is not uniform as per journal format.
AUTHORS: Thank you very much for your valuable suggestion. Thanks for reviewer’s comments; we modified all the references
